# Inter-Rater Reliability of Magnetic Resonance Imaging in Comparison to Computed Tomography and Wrist Arthroscopy in SLAC and SNAC Wrist

**DOI:** 10.3390/jcm10163592

**Published:** 2021-08-15

**Authors:** Athanasios Terzis, Arlena Klinger, Jessica Seegmüller, Michael Sauerbier

**Affiliations:** 1Department of Plastic, Hand and Reconstructive Surgery, BG Trauma Center Frankfurt am Main, Friedberger Landstrasse 430, 60389 Frankfurt am Main, Germany; arlena.klinger@web.de (A.K.); jessica.seegmueller@bgu-frankfurt.de (J.S.); 2Private Practice for Hand and Plastic Surgery, 61348 Bad Homburg v. d. Höhe, Germany; sauerbier@profsauerbier.com

**Keywords:** SLAC, SNAC, MRI, CT, wrist arthroscopy, cartilage

## Abstract

The aim of the study was to assess the inter-rater reliability of magnetic resonance imaging (MRI) in comparison to computed tomography (CT) and wrist arthroscopy in patients with scapholunate (SLAC) or scaphoid non-union advanced collapse (SNAC) as well as to evaluate a grading score of cartilage lesions. A total of 42 patients (36 male, 6 female) at a mean age of 45 years (range: 19–65 years) with a SLAC or SNAC wrist who had a preoperative MRI and CT scan as well as underwent arthroscopy of the wrist between 2013 and 2018 were included in this study. Cartilage lesions, as assessed by MRI, CT and wrist arthroscopy, were classified by two hand surgeons in three stages. Inter-rater reliability was evaluated using the Kendall Tau-b test as well as the chi-square test to analyze for trend. The correlation between cartilage lesions, classified by arthroscopy and MRI, was low. A moderate correlation between CT and arthroscopy staging was shown. The highest inter-rater correlation was found between MRI and CT staging. An additionally performed logistic regression showed that progression of cartilage lesions as shown in MRI scans correlates with a restriction of range of motion (ROM). The level of cartilage lesion may be more severely classified in an MRI than during arthroscopy. Arthroscopy remains the gold standard in detecting cartilage lesions and thus in the decision-making process of the definitive treatment in carpal collapse.

## 1. Introduction

Scaphoid non-unions as well as untreated injuries of the scapholunate ligament can alter the anatomy and biomechanics of the wrist and may develop into advanced carpal collapse [1,2,3,4]. During the decision-making process for definitive treatment, defining the exact stage of osteoarthritis plays a central role [5,6,7]. The gold standard for assessing the status of the cartilage is wrist arthroscopy [8]. In most cases, cartilage degeneration is classified arthroscopically according to the Outerbridge classification [9]. In addition, staging of cartilage degeneration may be assessed by magnetic resonance imaging (MRI) and/or computed tomography (CT). MRI is primarily intended to detect a scapholunate ligament tear [10,11] or to examine the blood perfusion of the scaphoid non-union [12]. It is a suitable method for mapping the hyaline cartilage of a joint [13]. However, there is no established MRI classification for degeneration of carpal cartilage [14].

The aim of this study was to evaluate the inter-rater reliability of the MRI in classifying cartilage lesions in patients with SNAC or SLAC wrist in comparison to the stage described by the CT scan and wrist arthroscopy.

Our hypothesis was that classification of cartilage lesions of the wrist by MRI alone is not as reliable as wrist arthroscopy.

## 2. Patients and Methods

After approval of the local ethics committee, 42 patients (36 male; 6 female) were included in this retrospective study between 2013 and 2018 at a mean age of 45 years (range: 19–65 years) with a SNAC or SLAC wrist, who had either an MRI (Avanto 1.5 T, Skyra 3 T, GE Healthcare Siemens, Erlangen, Germany) or CT scan (Sensation 128-Slice, GE Healthcare Siemens, Erlangen, Germany) in addition to wrist arthroscopy within a span of 6 months. The CT scan was considered optional and patients without preoperative CT scan were included, given that they had an MRI scan. Exclusion criteria were the presence of wrist panarthrosis as well as metal hardware in the wrist or hand. According to these criteria, an anonymized patient database was created. The data required was taken from the in-hospital discharge letters and surgical reports and then compiled in an anonymized form. The follow-up diagnosis of wrist arthroscopy was carried out with the help of arthroscopic recordings and surgical reports. The associated MRI or CT scans assessed the level of cartilage degeneration.

The level of cartilage degeneration in the scaphoid fossa was divided into three comparable stages in the arthroscopy as well as in the MRI and CT findings (Table 1). The staging was carried out by two independent hand surgeons and was determined by consensus.

Wrist arthroscopy was performed using a standardized protocol either in regional anesthesia or general anesthesia and using a tourniquet and wrist traction. The radiocarpal and midcarpal joints were examined with a 2.4 mm arthroscope (Karl Storz, Tuttlingen, Germany) through the 3/4 and MCR portals.

MRI scans were performed using the standardized protocol of our institution on either a 1.5 or 3 Tesla magnetic resonance imaging (Avanto 1.5 T, Skyra 3 T, GE Healthcare Siemens, Erlangen, Germany).

The data of 42 patients that underwent wrist arthroscopy and additionally had MRI and partially CT findings were evaluated by correlating the MRI, CT and arthroscopy findings on cartilage lesions in the scaphoid fossa according to the proposed classification (Table 1).

Statistical analysis was performed using the IBM SPSS-25 software (Armonk, NY, USA). The inter-rater correlation was determined to investigate the consistency of the assigned cartilage lesion in arthroscopy, MRI and CT. For this purpose, the Kendall Tau-b test was used, as a ranking correlation coefficient. Values above 0.6 are considered to be a moderate correlation; values above 0.8 indicate a very high correlation [15].

In addition, a chi-square test to evaluate the trend was conducted between the individual diagnostic methods. It was used in order to find out whether or not the defined MRI stages have a statistically significantly greater trend than the arthroscopy stages.

The correlation between cartilage lesion and range of motion (ROM) was assessed using a logistic regression and was shown in a scatter plot with a regression curve.

## 3. Results

A total of 42 patients were included in the study—36 male (85.71%) and 6 female (14.29%), at a mean age of 45.5 (19–65) years. Twelve patients (28.57%) suffered from SNAC wrist, 30 patients (71.42%) from SLAC wrist. A preoperative MRI scan was performed in all 42 patients, whilst a preoperative CT scan was performed in 26 patients (61.9%). Thirty patients (71.43%) received an MRI in an external practice or hospital; in 12 patients (28.57%), an MRI was performed in house. A total of 35.71% of the MRIs were performed with an intravenous contrast medium (gadolinium); the remaining 64.29% of them were performed without it.

The correlation between the described stages of cartilage lesion in wrist arthroscopy and MRI using the Kendall Tau-b test was 0.593, which can be considered to be a low match (Table 2). In addition, a chi-square test was performed to measure trend and find out if the defined MRI stages have a significantly greater trend than the defined arthroscopy stages. There was no statistically significant trend found (*p* = 0.1158), but graphically the MRI stages of cartilage lesions were more severe than the stages in arthroscopy.

A remarkable correlation between arthroscopy and CT findings was shown with a Kendall’s Tau-b of 0.692 (Table 3). In the chi-square test to evaluate trend, there was no significant trend found between arthroscopy and CT stages (*p* = 0.4211).

In the retrospective comparison between MRI and CT findings, the highest inter-rater correlation was found with a Kendall’s Tau-b value of 0.854 (Table 4). The chi-square test for trend showed a statistically significant trend (*p* = 0.047) between MRI and CT staging. However, there were more MRI than CT examination findings, which is a limitation of our study.

Furthermore, this study assessed whether or not progression of cartilage lesions as shown in an MRI correlates with a limitation of range of motion (ROM). The correlation was performed with a logistic regression and displayed in a simple scatter plot with a regression curve (Figure 1). The logistic regression showed that the MRI staging of cartilage lesions significantly correlates with ROM restriction (*p* = 0.027).

## 4. Discussion

The results of this study show that the MRI is a useful diagnostic tool for cartilage assessment in patients with SNAC or SLAC wrist. Furthermore, a standardized MRI classification of cartilage lesion could be helpful. In contrast to injuries of the bony structures of the wrist, which are easily diagnosed with native radiographs, advanced diagnostic procedures are required for detecting soft tissue injuries [16]. Exact staging of carpal collapse is essential for choosing the appropriate treatment [17,18,19]. With respect to healthcare efficiency, an accurate diagnosis is necessary for deciding between conservative or surgical regime and plays a crucial role in predicting treatment costs [19,20].

This study assessed the diagnostic possibilities of the MRI in detecting cartilage lesions in carpal collapse. As in other studies, wrist arthroscopy is used as a reference, which is the diagnostic gold standard for detecting cartilage lesions [8]. Arthroscopy is a minimally invasive surgical procedure with a very low complication risk. However, sufficient knowledge and surgical experience is required [21]. Therefore, cartilage assessment by means of an MRI scan could spare a surgical procedure prior to the definitive treatment. However, there is no established MRI classification for evaluating cartilage degeneration in carpal collapse [22].

The aim of this study was to validate the MRI as a predictive tool for cartilage lesion in carpal collapse. In patients with SNAC and SLAC wrist, cartilage lesions in the scaphoid fossa were classified in three stages according to arthroscopic, MRI and CT findings. Significant correlations were found between all three diagnostic methods. The inter-rater correlation between the described stages in arthroscopy and MRI showed a low match. A remarkable correlation was found between arthroscopy and CT findings. The highest inter-rater correlation was shown in the comparison between MRI and CT findings, whereby the CT examination was considered optional and thus was only performed in 26 patients. Hence, why only 26 findings could be compared to 42 MRI scans (Table 5).

Several studies have examined the significance of MR wrist arthrography to arthroscopy [23,24]. In contrast to the native MRI, MR arthrography is a semi-invasive procedure using an intra-articular contrast medium. Schmitt et al. showed in their study on 125 patients that MR wrist arthrography may detect lesions of the SL ligament and ulnocarpal complex but is inferior to arthroscopy when detecting pathologies of the hyaline cartilage and the LT ligament. In comparison to arthroscopy, MR arthrography had a sensitivity of 84.2% and a specificity of 96.2%in the diagnosis of cartilage lesions [13]. In their study on diagnostic comparison between MR arthrography and wrist arthroscopy Mahmood et al. have focused on SL ligament lesions, LT ligament lesions and TFCC lesions [23]. In summary, these studies show that although MR arthrography cannot yet replace arthroscopy, it is an adequate alternative for detecting ligament injuries of the wrist [25]. Although MR arthrography is described as a superior diagnostic method for the detection of intra-articular lesions of the wrist, its invasiveness involves risk and additional costs [24]. 

An important advantage of wrist arthroscopy is that it serves not only as a diagnostic but also as a therapeutic tool [23]. Haims et al. could show that the MRI does not have sufficient sensitivity to cartilage defects of the wrist [26]. However, despite the limited sensitivity to cartilage assessment of the wrist, the native MRI is still the least invasive procedure, which is why it should take the lead in standard wrist pain diagnostics. A study by Ochmann et al. confirmed that the 3 Tesla MRI examination with a hand coil is promising for cartilage assessment and has a much more accurate diagnosis compared to the 1.5 Tesla MRI examination [27,28]. In the assessment of cartilage lesions, a very good correlation between the 3 Tesla MRI and wrist arthroscopy was shown. Further studies are needed to determine whether or not the 3 Tesla MRI should be used as an alternative for diagnostic arthroscopy.

The hypothesis of this study was that cartilage lesions of the scaphoid fossa can be detected in the MRI and staged according to our proposed classification. It was shown that the stages of cartilage lesions using an MRI scan had a lower inter-rater correlation than wrist arthroscopy. Interestingly, there was a tendency to classify cartilage lesions seen in the MRI as more severe than in wrist arthroscopy. Although no statistical significance was demonstrated, the trend was evident in the chi square test. This means that the MRI staging classification may lead to a more severe false diagnosis. For example, in the case of a true stage 1 carpal collapse, a false diagnosis of a stage 2 carpal collapse could lead to a salvage operation, which would have never been initially recommended [17,29,30,31].

Due to the low correlation between MRI and arthroscopy staging, MRIs do not completely replace diagnostic arthroscopy. This statement was supported by an earlier study by Mutimer et al. [32]. The authors examined the comparability of the MRI and arthroscopy for the assessment of wrist cartilage. Only a moderate correlation was found between the two diagnostic methods, with the exclusive use of a 1.5 Tesla MRI device.

Our comparison between the MRI and CT stages showed a strong inter-rater correlation. The current literature describes that conventional CT scan as inferior to invasive CT arthrography in the assessment of articular cartilage. Furthermore, CT arthrography is partially superior to MR arthrography as a result of its higher resolution for cartilage assessment of small joints [33]. However, arthroscopy remains superior not only because of its better visualization and safe diagnosis, but also because of its direct therapeutic possibilities [23].

Most of the existing studies investigating the value of the CT scan for cartilage assessment were performed on the knee or hip [34]. Not much data is available for the diagnosis of advanced carpal collapse. Disadvantages of the CT scan over arthroscopy and MRI scan are radiation exposure and the fact that the cartilage surface is indirectly assessed, by evaluating the subsequent changes that occur after cartilage lesion [35]. Nevertheless, the CT scan is an alternative to the MRI in the diagnosis of advanced carpal collapse.

Furthermore, range of motion (ROM) of the wrist was evaluated according to our patient database. The early onset of osteoarthritis does not necessarily exhibit clinical symptoms [36,37]. Therefore, a further hypothesis of the study was that advanced stages of cartilage lesions correlate to a reduction of ROM. A significant correlation was shown here. The higher the level of cartilage degeneration shown in the MRI scan, the more limited the ROM of the wrist was. No significant association could be demonstrated between ROM and CT or the arthroscopy stages. Other parameters used were grip strength and pain level. However, a correlation analysis of these measurements was not possible as a result of the large inter-individual differences. There were no baseline values for grip strength and pain level was not documented in every case.

Our study has some limitations. Firstly, this was a retrospective study design. As a result, MRI and CT stages were known to the surgeon when performing wrist arthroscopy. Further limitations are the low patient number, the mixed patient collective and the limited evaluation of the documented data. In addition to data collection, the study was mainly based on the follow-up of appropriate MRI, CT and arthroscopy images. Not every MRI or CT scan was performed in our clinic as patients came to appointments with existing radiological examinations (71.43%). Furthermore, in this subgroup of patients with existing MRI scans, there were 15 cases of MRI scans performed with an intravenous contrast agent (gadolinium).

For a more reliable assessment of the MRI scan in detecting carpal cartilage damage in comparison to CT and arthroscopy, diagnostic evaluation should be performed with the same MRI device and evaluated by the same radiologist.

## 5. Conclusions

The current diagnostic gold standard for evaluation of cartilage status of the wrist is arthroscopy. Diagnostic arthroscopy is, however, a surgical procedure. An MRI scan is frequently performed as a diagnostic tool in carpal collapse. There is currently no established MRI classification for evaluating cartilage lesions. An evaluation of the cartilage in the MRI scan would be desirable because follow-up surgery could be avoided. A CT scan is another method of detecting cartilage lesions and remains an additive diagnostic tool for imaging in carpal collapse.

The aim of this study was to evaluate the diagnostic efficacy of the MRI in detecting cartilage lesions in patients with a SLAC or SNAC wrist. A retrospective follow-up data collection was used to determine whether or not cartilage lesions in carpal instability can be classified using an MRI and to compare the findings between wrist arthroscopy and CT scans.

Significant correlations were found between all three diagnostic techniques. The correlation between cartilage lesion stages in arthroscopy and MRI scans was low (Kendall-Tau-b 0.593). A moderate correlation was shown between arthroscopy and CT staging (Kendall-Tau-b 0.692). The highest correlation was found between the MRI and CT scans (Kendall-Tau-b 0.854).

The results of this study indicate that the classification of cartilage lesions in carpal collapse based on MRI scans alone is still insufficient because of its low correlation to arthroscopy staging A standardized MRI classification of cartilage lesions as well as a better availability of high-magnetic field MRI devices (3 Tesla) could help in the decision-making process for definitive treatment of carpal collapse. More studies using 3 Tesla MRI devices with a hand coil as well as a larger patient population would be needed to further investigate this question.

## Figures and Tables

**Figure 1 jcm-10-03592-f001:**
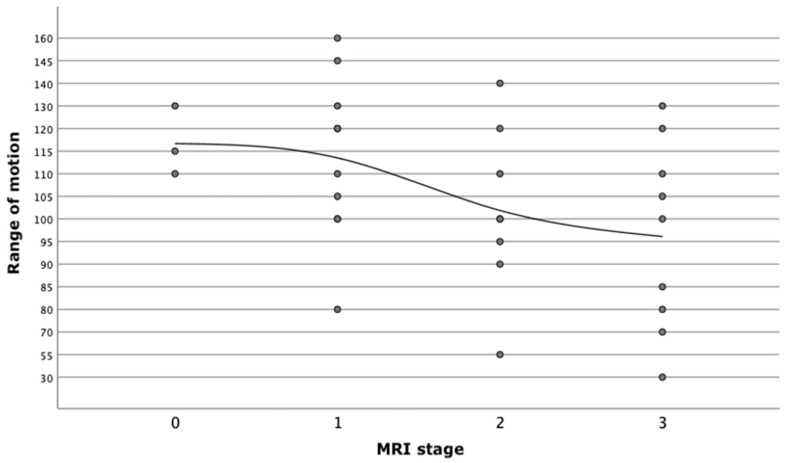
Correlation between MRI stage of cartilage lesion and restriction of range of motion (ROM).

**Table 1 jcm-10-03592-t001:** Classification of cartilage lesions in the scaphoid fossa.

Stage	MRI	CT	Arthroscopy
1	superficial cartilage lesion	joint narrowing	superficial cartilage lesion
2	deep cartilage lesion	subchondral sclerosis	deep cartilage loss
3	total cartilage loss	joint narrowing, cyst formation and osteophytes	total cartilage loss

**Table 2 jcm-10-03592-t002:** Correlation between cartilage lesion stage in MRI and arthroscopy.

			Arthroscopy	MRI
Kendall-Tau-b	arthroscopy	Correlation coefficient	1.000	0.593 *
		Sig. (2-sided)		0.000
		n	42	42
	MRI	Correlation coefficient	0.593 *	1.000
		Sig. (2-sided)	0.000	
		n	42	42

*: the correlation is on 0.01 significant (2-sided).

**Table 3 jcm-10-03592-t003:** Correlation between cartilage lesion stage in arthroscopy and CT.

			Arthroscopy	CT
Kendall-Tau-b	arthroscopy	Correlation coefficient	1.000	0.692 *
		Sig. (2-sided)		0.000
		n	26	26
	CT	Correlation coefficient	0.692 *	1.000
		Sig. (2-sided)	0.000	
		n	26	26

*: the correlation is on 0.01 significant (2-sided).

**Table 4 jcm-10-03592-t004:** Correlation between cartilage lesion stage in MRI and CT.

			MRI	CT
Kendall-Tau-b	MRI	Correlation coefficient	1.000	0.854 *
		Sig. (2-sided)		0.000
		n	26	26
	CT	Correlation coefficient	0.854 *	1.000
		Sig. (2-sided)	0.000	
		n	26	26

*: the correlation is on 0.01 significant (2-sided).

**Table 5 jcm-10-03592-t005:** Inter-rater correlation between cartilage legion stage in MRI, CT and arthroscopy.

			MRI	CT	Arthroscopy
Kendall-Tau-b	MRI	Correlation coefficient	1.000	0.854 *	0.593 *
		Sig. (2-sided)		0.000	0.000
		n	42	26	42
	CT	Correlation coefficient	0.854 *	1.000	0.692 *
		Sig. (2-sided)	0.000		0.000
		n	26	26	26
	arthroscopy	Correlation coefficient	0.593 *	0.692 *	1.000
		Sig. (2-sided)	0.000	0.000	
			42	26	42

*: the correlation is on 0.01 significant (2-sided).

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
