# Peer review of "Inter-Rater Reliability of Magnetic Resonance Imaging in Comparison to Computed Tomography and Wrist Arthroscopy in SLAC and SNAC Wrist"

_jcm, 2021, doi:10.3390/jcm10163592_

Round 1

Reviewer 1 Report

Very good evalution of the diagnostic possibilities of MRI in detecting cartilage lesion in patients with SLAC or SNAC wrist. A retrospective follow-up data collection was used to determine whether cartilage lesions in carpal instability can be classified on MRI and whether the assessment correlates with arthroscopy and CT findings. The manuscript is pertaining to the content and is written in fluent english. I would recommed to cite references in a chronological order as they appear in the text - please change.

Author Response

Dear reviewer,

thank you very much for your comments and suggestions. I will change the reference in a chronological order, as you recommend.

Reviewer 2 Report

This study is a primary research with  the retrospective study design, but I feel it is very interesting.

Only a few questions.

・P212 Why was the MRI classification  best correlated with the restriction of ROM.

・p254-254 leap in conclusion

・How do you use CT, MRI, and arthroscopy properly? Which is your first choice?

Author Response

Dear reviewer,

thank you very much for your comments and recommendations. I will revise the article with a native English-speaking colleague.

Comment 1: One of the hypotheses of this study was that MRI may detect cartilage lesions that correlate with restriction of ROM. In this way, MRI would be an early predictive tool for reduced hand function due to carpal collapse. Cartilage is best depicted in MRI scan, compared to CT scan. When compared to CT and arthroscopy findings, only MRI correlated significantly to ROM restriction, thus showing that severe cartilage lesions may lead to loss of hand function.

Comment 2: You are absolutely right, I will remove the leap in the conclusion.

Comment 3: Depending on the aetiology of carpal collapse, we always perform a CT scan in SNAC wrist and an MRI scan in SLAC wrist prior to wrist arthroscopy. We always perform a diagnostic wrist arthroscopy in order to assess the exact stage of collapse. Only wrist arthroscopy allows for setting the precise diagnosis and is the most helpful tool in decision making.